# The Hidden Hunger among Nepalese Non-Pregnant Women Aged 15–49 Years: The Role of Individual, Household, and Community-Level Factors

**DOI:** 10.3390/ijerph21070875

**Published:** 2024-07-04

**Authors:** Kingsley Emwinyore Agho, Stanley Chitekwe, Biniyam Sahiledengle, Lucy Ngaihbanglovi Pachuau, Sanjay Rijal, Naveen Paudyal, Sanjeev Kumar Sahani, Andre Renzaho

**Affiliations:** 1School of Health Sciences, Western Sydney University, Campbelltown, NSW 2560, Australia; l.ngaihbanglovi@westernsydney.edu.au; 2Translational Health Research Institute (THRI), Western Sydney University, Penrith, NSW 2560, Australia; andre.renzaho@westernsydney.edu.au; 3Faculty of Health Sciences, University of Johannesburg, Johannesburg 2094, South Africa; 4Nutrition Section, United Nations Children’s Fund (UNICEF) Ethiopia, Addis Ababa 1169, Ethiopia; schitekwe@unicef.org; 5Department of Public Health, Madda Walabu University Goba Referral Hospital, Bale-Goba 4540, Ethiopia; biniyam.sahiledengle@gmail.com; 6United Nations Children’s Fund (UNICEF), Nepal Country Office P.O. Box 1187, United Nations (UN) House, Pulchowk, Kathmandu 44600, Nepal; sarijal@unicef.org (S.R.); npaudyal@unicef.org (N.P.); ssahani@unicef.org (S.K.S.)

**Keywords:** micronutrients, women, deficiencies, non-pregnant women, Nepal

## Abstract

Micronutrient deficiencies remain a public health burden among non-pregnant women in developing countries, including Nepal. Hence, this study examined micronutrient deficiencies among non-pregnant Nepalese women aged 15–49 using the 2016 Nepal National Micronutrient Status Survey (NNMSS). Data for 2143 non-pregnant women was extracted from the 2016 NNMSS. The study analysed the levels of ferritin, soluble transferrin receptor (sTfR), red blood cell (RBC) folate, and zinc of the participants. Multivariable logistic analysis was carried out to assess factors associated with micronutrient deficiencies. The prevalence of ferritin, sTfR, folate, and zinc was observed to be 19%, 13%, 16%, and 21%, respectively. Non-pregnant women from the Janajati region were significantly less prone to high levels of ferritin [adjusted odds ratio (AOR): 0.45; 95% confidence interval (CI): 0.25, 0.80], and those who had body mass index (BMI) of 25 kg/m^2^ or higher had significantly elevated ferritin levels [AOR: 2.69; 95% CI: 1.01, 7.17]. Non-pregnant women aged 35–49 years were significantly less predisposed to folate deficiency [AOR: 0.58; 95% CI: 0.40, 0.83], and the odds of zinc deficiency were significantly lower among non-pregnant women from wealthier households [AOR: 0.48; 95% CI: 0.31, 0.76]. This study provides further insight into screening high-risk subgroups and instituting public health interventions to address the prevailing micronutrient deficiencies among non-pregnant Nepalese women.

## 1. Introduction

Micronutrient deficiency arises from insufficiencies of minerals and soluble vitamins [1]. The most common micronutrient deficiencies include reduced levels of folate, vitamin A, iron, iodine, and zinc. These deficiencies may adversely affect an individual’s physical, mental, and cognitive development [2,3]. There is evidence that an estimated two billion people suffer from deficiencies in vital micronutrients [4,5]. Globally, nutritional iron deficiency (which may cause anaemia, reduced working and learning capacity, increased maternal and infant mortality, low birth weight, and impaired function at all stages of human life [6]) seems to be the most prevalent among the micronutrient deficiencies [7,8]. 

Hidden hunger, also known as multiple micronutrient deficiencies, such as those involving iron, zinc, iodine, and vitamin A, can occur even with an adequate energy intake due to consuming a diet that is rich in energy but lacking in nutrients [9]. Nutritional iron deficiency is said to arise when physiological requirements are not met by the absorption of iron from the diet [10]. Furthermore, dietary iron bioavailability has been found to be low in populations where the consumption of plant-based diets is unvarying [10]. The status of micronutrients varies significantly between populations and throughout pregnancy. Nutritional iron deficiency affects all populations worldwide and is most common in developing countries [11,12]; this condition is accompanied by substantial health and economic costs, which may include poor pregnancy outcomes, impaired academic performance among school children, and a decrease in productivity [10]. Zinc plays a key role in the catalytic activity of about 100 enzymes [13], and is also vital for immune function [14,15,16], the synthesis of protein [15], wound healing [17], and to support normal pregnancy growth and development, as well as normal growth during childhood and adolescence [18]. Additionally, a zinc shortage has been linked to impaired cognition and behaviour [16]. Zinc deficiency is characterised by loss of appetite, hair loss, diarrhoea, delayed sexual maturation, male impotence, and eye and skin lesions [19,20]. Folate is obtained by consuming foods high in dietary folate, such as green leafy vegetables, legumes, and fortified cereals. Individuals with folate deficiencies are more likely to exhibit neural tube and other congenital disabilities, heart disease, stroke, and depression [21]. 

The extant literature is replete with studies on micronutrient deficiencies in Nepal [6,22,23,24,25,26,27]. However, whilst some studies merely examined the prevalence and individual, household and community factors associated with micronutrient deficiencies [6,22], other studies examining the individual, household and community factors associated with micronutrient deficiencies were limited in scope; for instance, one study assessed the micronutrient deficiencies among adolescents, but the studies were limited to refugees [24]. Other studies also focus on children aged 6–59 months [28,29]. Although micronutrient deficiencies significantly affect women’s overall well-being and health [30], little is known about the socio-demographic and dietary habits contributing to micronutrient deficiencies in non-pregnant Nepalese women. The current study utilised data from the Nepal National Micronutrient Status Survey (NNMSS) to examine the association of iron, folate, and zinc deficiencies with individual-, household-, and community-level factors, as well as health status and dietary habit factors among non-pregnant women in Nepal. By identifying the significant factors associated with these deficiencies, our study can inform the government of Nepal, public health practitioners and other stakeholders, enabling them to plan more effective interventions for micronutrient deficiency alleviation in Nepal by targeting the most vulnerable populations.

## 2. Materials and Methods

### 2.1. Data Source, Design, and Participants

The current study utilised data from the 2016 Nepal National Micronutrient Status Survey (NNMSS). The NNMSS, which is cross-sectional, population-based, and nationally representative in design, was stratified to showcase representative estimates for Nepal’s five development regions: Eastern, Central, West, Midwest, and Far West, and the three ecological zones of Nepal: the terai (plains), hills, and mountains. Details of the NNMSS can be obtained from the extant literature [31]. We collected nationally representative samples of adolescent non-pregnant Nepalese women aged between 15 and 49 years of age. 

### 2.2. Data Collection

Information on each household and for the non-pregnant women aged 15–49 years was collected using administered questionnaires. The household questionnaire was used to identify all household members eligible to participate in the survey (i.e., one of the target groups). The questionnaire for the non-pregnant women was used to collect information regarding socio-demographic characteristics, participation in key national nutrition and other interventions, recent micronutrient supplementation intake (zinc, iron, folic acid, multiple micronutrients) and if consumed within the last 24 h, timing since most recent intake for zinc supplementation; types of food groups consumed by women 15–49 years were collected, based on the intake of the preceding day. These food groups included animal-source foods, as well as dark leafy greens, along with a two-week recall of incidence of fever, cough, and diarrhoea; all this information was collected through questionnaires. Anthropometric measures, such as weight and height, were collected as reported elsewhere [32]. Informed consent was obtained from the respondents prior to their participation. Written informed consent was obtained from those who were educated and could read, while oral informed consent, with witness signatures, was obtained from those who could not read. Full details regarding consent from participants were reported elsewhere [32]. 

### 2.3. Outcome Indicators

The Biomarkers Reflecting Inflammation and Nutrition Determinants of Anemia (BRINDA) working group recommends adjusting serum zinc, sTfR, and ferritin requirements for non-pregnant women aged 15–49 years, and these recommendations were used in this study [33]. In the current study, the variables outcome included deficiencies in iron ferritin and sTfR concentrations, along with folate and zinc. The assessment of iron/ferritin deficiency covered all five regions and the three ecological zones. The World Health Organisation (WHO)’s recommended indicator of iron deficiency in populations is the level of ferritin, which measures iron stores in the body and is an acute-phase reactant protein influenced by inflammation and infections. Ferritin is inexpensively analysed using the enzyme-linked immunosorbent assay (ELISA) method [32]. An indicator of iron insufficiency and that iron stores are depleted (assuming the absence of other causes of abnormal erythropoiesis) is known as the presence of soluble transferrin receptor (sTfR), which can be elevated by thalassemia and is thought to be less influenced by inflammation and infection than ferritin. The recommended analysis volume for ferritin and sTfR is 30 μL [34]. To analyse retinol-binding protein (RBT), the recommended volume is 30 μL. The microbiological assay was the gold standard for analysing RBC folate [35]. The recommended volume for analysing RBC folate was 15 μL. Iron deficiency was defined as adjusted ferritin levels < 15.0 μg/L. Zinc deficiency was classified as zinc < 66.0 μg/dL for non-fasted morning samples and <59.0 μg/dL for non-fasted afternoon samples for non-pregnant 15–49-year-old women [36,37]. Detailed information regarding the classification of serum zinc, sTfR, ferritin, RBC folate, and Vitamin A has been described elsewhere [32,33]. 

### 2.4. Potential Covariates

Past studies, as well as the available factors in the NNMSS, informed our choice of potential confounding factors for this study on micronutrient deficiency [38,39,40,41,42]. We grouped the potential confounding factors into individual, household, and community-level factors, as well as anthropometric characteristics, health status, and dietary habits (a day prior to the survey). 

The individual-level factors included the participants’ education, age, and marital status. The household-level factors comprised their ethnicity and household wealth index. We applied the principal components analysis (PCA) method to the household wealth index [43] as a score regarding household assets. After computing the index, each participant household was ranked by its score. The wealth index was categorised into five quintiles: the bottom 20% was classified as poorest, the next 20% was classified as poorer, the next 20% was classified as middle, the next 20% was classified as richer, and the top 20% was classified as richest [43]. The mean weight, height, body iron stored (BIS) (obtained from the ferritin test used to measure the amount of iron stored in the body) [33], and body mass index (BMI) constituted the anthropometric factors; health status factors were made up of contraction of fever, cough and diarrhoea over the two weeks preceding the survey; recent micronutrient supplementation intake related to folic acid supplements and multivitamins consumed within the previous 24 h.

The community-level factors consisted of place of residence, state, geographical region, and ecological zone (terai, hills, and mountain), and the food consumed variables, which comprise the diet habit factor, consist of the consumption of the different food groups that contain animal-source foods and dark leafy greens.

### 2.5. Statistical Analysis

This study’s statistical analyses were conducted using STATA/MP version 14 (Stata Corp, College Station, TX, USA). To allow for adjustments for the cluster-sampling design and weight, we used the ‘*Svy*’ commands [44]. We first executed frequency tabulations to describe the data used in the study, as well as the proportions of iron, folate, and zinc deficiencies. Only those that produced a proportion >4% were used to determine associated factors because a small prevalence (<4%) would result in wider confidence intervals and biased conclusions. Next, we employed survey mean analyses to calculate the lower quartile (<25th percentile), middle quartile (≥25th percentile and ≤75th percentile), and upper quartile (baseline plasma values >75th percentile). The Taylor series linearisation method was utilised to estimate confidence intervals (CIs) for the prevalence estimates of the micronutrient deficiencies featured in this study. Additionally, the bivariable and multivariable logistic regressions that adjusted for clustering and sampling weights to determine the odds ratios of the deficiencies of the micronutrients were calculated. We then used a six-stage model as part of the multivariable logistic regression analysis to calculate the adjusted odds ratios of micronutrient deficiencies, and the rationale for choosing factors to be included was based on the cut-off point of *p*-value = 0.20, which is a good choice for the predictive model and an argument for the backward elimination process [45].

In the stage model, the individual-level factors were entered into the first model, and a manually executed elimination procedure to assess factors associated with micronutrient deficiencies at a 0.05 significance level was carried out. In the second model, the significant factors in the first stage were added to the household-level factors, followed by the elimination process. A similar approach was used for the community-level, anthropometric, health status, and diet habits factors in the third, fourth, fifth, and sixth stages.

In the final model, any co-linearity was tested for and reported. The odds ratios with 95% confidence intervals were then calculated to determine the adjusted risk of independent (possible confounding) variables.

## 3. Results

### 3.1. Characteristics of the Sample

The characteristics of the study participants are presented in Table 1. The study sample consisted of 2143 non-pregnant women aged between 15 and 49 years. The majority of the participants had no schooling (48%) and came from the Brahmi/Chettri caste group. The mean weight and height of the participants were 51.3 ± 9.9 kg and 151.4 ± 6.0 cm, respectively. The mean body iron stores (BIS) were 4.9 ± 3.8 mg/kg, and the majority of the participants (67%) had a BMI of between 19 and 25 kg/m^2^. About one-third of the non-pregnant women received iron and folic acid (IFA) supplements, while only 2% of them received any multivitamins from the starchy foods, legumes, and nuts which most participants consumed; a large majority of the participants did not consume any other food groups.

### 3.2. Prevalence of Micronutrient Deficiencies

The serum concentrations of the micronutrients assessed in the current study are shown in Table 2. The mean serum concentration of iron (ferritin biomarker) was 38.9 ± 27.5 μg/L, and the mean serum concentration of folate was 593.3 ± 308.5 nmol/L. Proportions and 95% confidence intervals of deficiencies in the micronutrients are shown in Figure 1. The prevalence of deficiencies in the featured micronutrients was 19%, 13%, 16%, and 21% for ferritin and sTfR iron concentrations, folate, and zinc, respectively.

### 3.3. Factors Associated with Micronutrient Deficiencies

The odds of being iron deficient (according to sTfR concentrations) in non-pregnant women from the Janajati ethnic group were significantly lower compared with those for the Brahmin/Chettri ethnic group [AOR: 0.45; 95% CI: (0.25, 0.80)] (Table 3). However, non-pregnant women from the same ethnic group were significantly more likely to exhibit iron deficiency (according to sTfR levels) compared with those from the Brahmi region [AOR: 2.42; 95% CI: (1.39, 4.21)].

The odds of iron deficiency (ferritin concentrations) were significantly higher among non-pregnant women whose BMI was ≥25 kg/m^2^, compared with those whose BMI was ≤18.5 kg/m^2^ [AOR: 2.69; 95% CI: (1.01, 7.17)] (Table 3).

Non-pregnant women from ethnic (other Terai caste, Newar, and Muslim) groups other than the Janajati and Dalit castes were significantly more predisposed to folate deficiency compared with those from the Brahmin ethnic group [AOR: 2.79; 95% CI: (1.69, 4.61)] (Table 3). Furthermore, non-pregnant women from the Far-Western region were significantly more likely to be folate deficient compared with those from the Eastern region [AOR: 3.10; 95% CI: (1.48, 6.49)]. Non-pregnant women aged 25–34 years were significantly less prone to folate deficiency compared with those aged 15–24 years [AOR: 0.63; 95% CI: (0.48, 0.83)]. The likelihood of folate deficiency was significantly lower among non-pregnant women from the richest households compared with those from the poorest households [AOR: 0.55; 95% CI: (0.30, 0.99)]. Non-pregnant women who consumed vitamin A-rich foods were significantly more prone to folate deficiency compared with those who did not [AOR: 2.34; 95% CI: (1.52, 3.61)] (Table 3).

The odds of zinc deficiency among non-pregnant women from richer households were significantly lower compared to those from the poorest households [AOR: 0.48; 95% CI: (0.31, 0.76)]. The odds of zinc deficiency were also significantly higher among non-pregnant women aged 35–49 years compared with those aged 15–24 years [AOR: 1.39; 95% CI: (1.01, 1.93)].

## 4. Discussion

In the current study, we identified factors associated with deficiencies in some micronutrients (i.e., ferritin concentrations, sTfR concentrations, folate, and zinc) among non-pregnant Nepalese women aged 15–49 years. Some of the factors included ethnic differences, regional differences, non-consumption of folic acid (IFA) supplements, high BMI (sTfR concentrations), low BMI (ferritin concentrations), non-consumption of some food groups such as meats and fish, and household poverty.

The risk of iron deficiency (according to ferritin concentrations) among non-pregnant women from the Janajati ethnic group of Nepal was found to be significantly less compared with that of those from the Brahmin ethnic group. Another study conducted in Nepal also found increased odds of iron deficiency (according to ferritin concentration) among children belonging to the Brahmin ethnic group [29]. However, for iron deficiency among this group of Nepalese women with sTfR as a biomarker, those from the Janajati ethnic group were significantly more prone to deficiencies compared with their Brahmin counterparts. In Benin, women from the Boo ethnic group exhibited a risk factor for iron deficiency [42]. This finding highlights differences by ethnicity in regards to the odds of iron deficiency, and it is a significant indicator of unequal distribution in nutrition in Nepal. There is, therefore, a need for further research on micronutrient deficiencies among and within various ethnic settings of Nepal. Other previous studies revealed that some sections of a country may be associated with iron deficiency, and such findings have often been found to be higher in rural areas compared to urban locations [46,47]. We also found that non-pregnant women from the Far-Western region had significantly higher odds of being folate deficient than their counterparts from the Eastern region. According to the Human Development Index and Human Poverty Index [47], the Far-Western region of Nepal is the most underdeveloped region, with low land productivity and poor access to health and education. It is a far distance from the centre of Kathmandu [48]. The practice of gender-based food distribution discrimination, discrimination during the menstruation cycle, employment constraints for women, and lack of land ownership all have tremendous impacts on food availability for women in this region [48,49]. Further research should be conducted to assess the differences and understand why non-pregnant women in some geographical regions are more prone to micronutrient deficiencies than those from other ethnic settings. Suitable interventions should target those disadvantaged ethnic groups to improve the burden of micronutrient deficiencies they carry.

In multivariable analysis, compared to non-pregnant underweight women (BMI < 18.5 kg/m^2^), significantly higher odds for the occurrence of iron deficiency were observed among overweight non-pregnant women (BMI ≥ 25 kg/m^2^). These findings could be attributed to the fact that being overweight as a non-pregnant women can lead to iron deficiency by inhibiting the absorption of dietary iron from the duodenum [50]. This can worsen the problem of micronutrient deficiencies, especially in populations where plant-based diets are common and the availability of dietary iron is already low [51]. This finding is also consistent with those of previous studies, which found that some adolescents who were at risk of being overweight or who were currently overweight were almost twice as likely to experience iron deficiency compared with those with normal weight [52,53]. Previous studies have also shown that BMI is inversely associated with anaemia, which is mostly caused by iron deficiency [39,54,55]. The findings show a similar relationship between iron deficiency and BMI among adolescents and non-pregnant women.

Our study found a significant correlation between consuming vitamin A-rich fruits and RBC folate levels in non-pregnant women. A cross-sectional study carried out in Malaysia supported this finding. The study focused on the impact of 24 h dietary recall on blood folate and showed that females reported higher folate intake than males. This finding could be attributed to their better eating habits and access to high-folate foods like fruits and vegetables [56].

Our study showed that non-pregnant women from the poorest households were significantly more predisposed to folate deficiency than their counterparts from the richest households. This finding was consistent with an observation made in a previous study in Japan, indicating that dietary intake of folate was significantly associated with socioeconomic status among Japanese workers [57]. Several other previous studies obtained similar findings [58,59,60,61]. Furthermore, we found that non-pregnant women from the poorest households were more likely to be deficient in zinc. Foods rich in zinc include red meat and poultry. Only individuals from relatively richer families can afford foods that are likely to contain zinc. Additionally, our study found that women from poorer households reported higher odds of zinc deficiency. This confirms our findings that deficiencies in folate and zinc were associated with household poverty. Our finding is consistent with those of studies conducted in Nepal, as well as many low-income and middle-income countries, which found a direct relationship between folate deficiency and household poverty [6,62,63,64]. This finding is not surprising, as people would need money to purchase folic acid supplements to boost their folate levels.

We found that non-pregnant women aged 35–49 years were significantly less prone to folate deficiency compared with their counterparts aged 15–24 years, which seemed to suggest that younger individuals were at a higher risk of folate deficiency compared to older ones. This finding seemed to be contrary to the findings of a previous study, which found that folate deficiency was strongly related to age, especially in over 35 years, and could also be attributed to reduced dietary intake and intestinal malabsorption [65]. However, we found that non-pregnant women aged 35–49 years had significantly higher odds of exhibiting zinc deficiency compared to those aged between 15 and 24 years, indicating a positive association between zinc deficiency and age. This aligns with previous research in Norway, which found an increased prevalence of zinc deficiency with age [66]. The positive correlation between increased age and zinc deficiency may be attributed to a decrease in the “absorptive capacity of the small bowel and a general decline in energy and intake of food among the aged” [67]. Further research is needed to examine how the age of non-pregnant women relates to folate and zinc deficiencies.

Our study also revealed that non-pregnant women who had a fever before the survey were significantly less predisposed to zinc deficiency. We could not find any study suggesting the association between the contraction of fever and zinc deficiency. Still, this occurrence could be due to health problems, including fever, faced by rural women, and a sub-analysis from this dataset using cross-tabulation that compared results between the place of residence and fever over the two weeks preceding the survey revealed that 9 in 10 non-pregnant women in rural areas had a fever compared to 1 in 10 in urban areas. However, previous studies have identified other diseases associated with zinc deficiency, including malabsorption syndrome, chronic liver disease, chronic renal disease, sickle cell disease, diabetes, and other chronic diseases [68]. Our study found that iron deficiency was associated with decreased iron stores in the body. This finding was supported by a study conducted by Soares et al. (2010) [68], which found that low levels of iron in the body result in reduced levels of iron in the blood, and further analysis of this study data revealed that the mean of iron stores in the body was significantly higher in older women (35–49 years) than younger women (15–24 years) (5.3 vs. 4.4, *p* < 0.001), respectively.

One of the main strengths of thecurrent study was that it utilised datasets from the NNMSS, which is population-based and nationally representative. Consequently, the study’s findings may be generalisable to any part of Nepal. Nonetheless, the current study is not without limitations. Most importantly, the study was cross-sectional in design; therefore, causation between these deficiencies (iron, zinc and folate) and associated factors could not be established, and future research using a longitudinal design, or an interventional study, is necessary to establish causality between these deficiencies (iron, zinc, and folate) and other associated factors. Recall bias may also have influenced our findings because self-reported data were collected and analysed. Consistent measures such as self-reports and combinations of technologies, including various sensor- and image-based tools for detecting solid and liquid dietary consumption, could reduce recall bias [69]. Furthermore, due to the high cost of collecting and storing blood for a nationally representative micronutrient survey, further research on dietary data and iron deficiencies (ferritin and sTfR concentrations) is important, and this will better prepare the Nepalese government and other stakeholders with the necessary tools to implement effective public health interventions when the opportunity arises.

## 5. Conclusions

Factors associated with some micronutrient deficiencies in non-pregnant Nepalese women were revealed in this current study. The factors included being from the Janajati ethnic group, low BMI, non-consumption of IFA supplements, geographic region and household poverty. These findings provide the Nepalese government and other stakeholders with the tools required to establish appropriate public health interventions to minimise the micronutrient deficiency burden among Nepal’s non-pregnant women. Particular attention should be paid to the Janajati ethnic group, poor households, and public health programmes dealing with the IFA supplementation supply. Other factors of vulnerabilty that should be targeted in interventional programmes should include non-pregnant women from the Far-Western region, as well as those in the youngest age bracket, 15–24 years.

## Figures and Tables

**Figure 1 ijerph-21-00875-f001:**
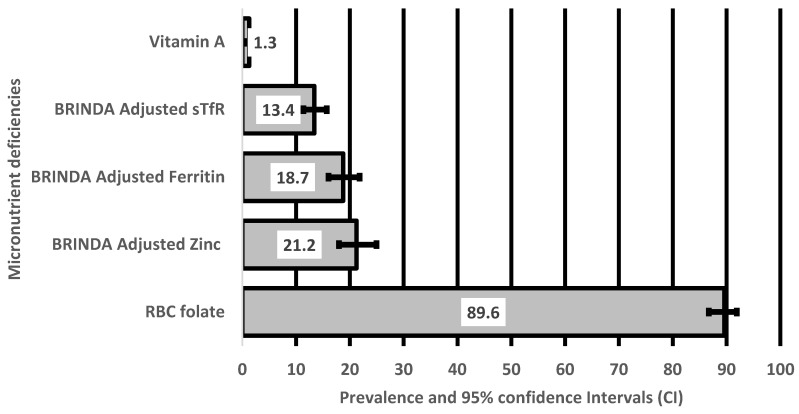
Prevalence and 95% confidence intervals of micronutrient deficiencies in non-pregnant women in Nepal.

**Table 1 ijerph-21-00875-t001:** Characteristics of non-pregnant Nepalese women aged 15–49 years (n = 2143).

Characteristics	%
Individual-level factors	
Age (years)	
15–24	31.2
25–34	35.9
35–49	35.9
Level of education completed	
No schooling	47.7
Completed primary	26.0
Secondary or more	26.3
Marital status	
Married	84.4
Not married	15.6
Household-level factors	
Household wealth index	
Poorest	15.4
Poorer	18.6
Middle	20.2
Richer	20.6
Richest	25.2
Ethnicity (Caste)	
Brahmin/Chettri	36.8
Dalit	15.4
Janajati	31.1
Others ^$^	16.7
Anthropometry	
Mean weight (SD)	9.9
Mean height (SD)	6.0
BIS (mg/kg)	
Mean BIS (SD)	3.8
BMI (kg/m^2^)	
≤18.5	9.9
19–25	67.0
25+	23.1
Health status	
Fever	13.8
Cough	14.6
Diarrhoea	9.2
Supplementation	
Received IFA supplements	
Yes	33.5
No	66.5
Received multivitamins	
Yes	2.0
No	98.0
Community-level factors	
Residence	
Urban	13.7
Rural	86.3
State	
Province 1	17.4
Province 2	20.3
Province 3	20.2
Province 4	10.1
Province 5	17.6
Province 6	4.9
Province 7	9.4
Geographical region	
Eastern	22.6
Central	35.3
Western	19.5
Mid-Western	13.2
Far-Western	9.4
Ecological zone	
Mountain	6.3
Hill	42.2
Terai	50.5
Food consumed a day before the survey	
Tubers/starchy foods (yes)	85.5
Legumes and nuts (yes)	74.0
Nuts and seeds or any food from nuts or seeds (yes)	6.0
Dairy products (yes)	43.6
Eggs (yes)	10.1
Liver, kidney, heart, or other organ meat or blood (yes)	6.9
Other meats (chicken, goat, buffalo, pigs, ducks) (yes)	25.9
Fish (fresh or dried fish, shellfish, prawn, crab, etc.) (yes)	5.5
Consumed other vitamin A-rich fruits (yes)	49.4
Vitamin A-rich fruits (yes)	10.5
Other fruits (yes)	81.7
Dark green leafy vegetables (yes)	13.7
Other vegetables (yes)	34.9
Mean food consumed (sd)	4.5 (1.7)

SD: standard deviation; BIS: body iron stores; IFA: iron and folic acid; $ = Other Terai Caste, Newar, and Muslim.

**Table 2 ijerph-21-00875-t002:** Cut-off values and levels of Vitamin A, RBC folate, BRINDA ferritin, sTfR and zinc deficiencies among pregnant women aged 15–49 years in Nepal.

Micronutrient	Mean (sd)	Quartiles			Deficiency *	
		Q25	Q50	Q75	n	%
Iron deficiency/Ferritin (n = 2135)	38.9 (27.5)	18.6	32.4	53.1	400	18.7
Iron deficiency/sTfR (n = 2135)	6.7 (4.6)	4.6	5.5	6.9	286	13.4
Folic acid/RBC folate (n = 2140)	593.3 (308.5)	405.0	549.1	726.7	340	15.9
Zinc (n = 2135)	84.5 (35.8)	62.5	80.6	99.2	453	21.2

*: Ferritin (<15 μg/L); sTfR (>8.3 mg/L); RBC folate (<305 nmol/L) and Zinc (<60 μg/dL or <66.0 μg/dL for non-fasted morning samples).

**Table 3 ijerph-21-00875-t003:** Factors associated with iron (ferritin and sTfR levels), RBC folate, and zinc deficiencies among non-pregnant Nepalese women aged 15–49 years.

Characteristic	Unadjusted OR (95% CI)	*p*-Value	Adjusted OR (95% CI)	*p*-Value
Iron deficiency (ferritin as a marker)				
Ethnicity (Caste)				
Brahmin/Chettri	1.00		1.00	
Dalit	0.72 (0.49, 1.09)	0.116	0.45 (0.20, 0.98)	0.044
Janajati	0.59 (0.43, 0.82)	0.002	0.45 (0.25, 0.80)	0.007
Others ^$^	0.96 (0.68, 1.35)	0.819	0.73 (0.34, 1.57)	0.416
BMI (kg/m^2^)				
≤18.5	1.00		1.00	
19–25	1.27 (0.78, 2.05)	0.333	2.12 (0.74, 6.08)	0.162
25+	1.05 (0.64, 1.74)	0.833	2.69 (1.01, 7.17)	0.048
BIS (mg/kg) body weight (continuous)	0.29 (0.23, 0.36)	<0.001	0.27 (0.21, 0.34)	<0.001
Height (in cm)	0.99 (0.97, 1.01)	0.103	0.96 (0.92, 0.99)	0.022
Administered any IFA supplements				
No	1.00		1.00	
Yes	0.72 (0.55, 0.94)	0.019	0.39 (0.21, 0.71)	0.002
Iron deficiency (sTfR as a marker)				
Ethnicity (Caste)				
Brahmin/Chettri	1.00		1.00	
Dalit	0.89 (0.59, 1.34)	0.561	1.08 (0.63, 1.86)	0.764
Janajati	1.23 (0.82, 1.84)	0.303	2.42 (1.39, 4.21)	0.002
Others ^$^	1.18 (0.62, 2.23)	0.614	1.15 (0.51, 2.57)	0.731
BIS (mg/kg) body weight (continuous)	0.58 (0.53, 0.62)	<0.001	0.56 (0.52, 0.60)	<0.001
BMI (kg/m^2^)				
≤18.5	1.00		1.00	
19–25	0.73 (0.49, 1.08)	0.118	0.49 (0.27, 0.91)	0.025
25+	0.79 (0.49, 1.29)	0.345	0.95 (0.48, 1.86)	0.87
Folic acid/RBC folate				
Ethnicity (Caste)				
Brahmin/Chettri	1.00		1.00	
Dalit	1.63 (0.99, 2.65)	0.051	1.44 (0.89, 2.34)	0.135
Janajati	1.02 (0.70, 1.48)	0.936	1.00 (0.66, 1.51)	0.997
Others ^$^	2.12 (1.29, 3.49)	0.003	2.79 (1.69, 4.61)	<0.001
Geographical region				
Eastern	1.00		1.00	
Central	1.34 (0.60, 3.01)	0.472	1.23 (0.59, 2.56)	0.569
Western	1.34 (0.62, 2.87)	0.451	1.37 (0.64, 2.92)	0.408
Mid-Western	2.34 (1.23, 4.44)	0.01	2.21 (1.10, 4.46)	0.027
Far-Western	2.99 (1.52, 5.87)	0.002	3.10 (1.48, 6.49)	0.003
Age (years)				
15–24	1.00		1.00	
25–34	0.62 (0.47, 0.82)	0.001	0.63 (0.48,0.83)	0.001
35–49	0.56 (0.39, 0.82)	0.003	0.58 (0.40, 0.83)	0.004
Household wealth index				
Poorest	1.00		1.00	
Poorer	0.71 (0.45, 1.13)	0.145	0.85 (0.57, 1.27)	0.431
Middle	0.73 (0.47, 1.15)	0.172	0.85 (0.55, 1.30)	0.446
Richer	0.63 (0.40, 0.99)	0.045	0.71 (0.49, 1.02)	0.062
Richest	0.40 (0.21, 0.77)	0.007	0.55 (0.30, 0.99)	0.048
Consumed Vitamin A-rich fruits				
No	1.00		1.00	
Yes	2.43 (1.47, 4.00)	0.001	2.34 (1.52, 3.61)	<0.001
Zinc				
Household wealth index				
Poorest	1.00		1.00	
Poorer	0.67 (0.43, 1.04)	0.072	0.66 (0.42, 1.03)	0.066
Middle	0.72 (0.50, 1.03)	0.074	0.73 (0.51, 1.06)	0.096
Richer	0.47 (0.31, 0.73)	0.001	0.48 (0.31, 0.76)	0.002
Richest	0.60 (0.40, 0.91)	0.016	0.61 (0.42, 0.91)	0.016
Age (years)				
15–24	1.00		1.00	
25–34	0.99 (0.72, 1.38)	0.973	1.00 (0.72, 1.38)	0.997
35–49	1.37 (0.98, 1.92)	0.068	1.39 (1.01, 1.93)	0.048
Exhibited fever				
No	1.00		1.00	
Yes	0.69 (0.51, 0.94)	0.018	0.69 (0.51, 0.94)	0.018

Potential adjusted covariates are individual, household and community factors, as reflected in Table 1. OR: odds ratio; CI: confidence interval; RBC: red blood cells; $ = Other Terai Caste, Newar, and Muslim, BIS: body iron stores, IFA: iron and folic acid.

## Data Availability

The study was conducted by analysing existing survey datasets available by applying to UNICEF Nepal, with all identifying information removed. Since this was a secondary data analysis and did not involve any interaction with the participants, written informed consent for the present analysis was not necessary. The data collection methods used in this analysis, including the consent process, were previously described in the Nepal National Micronutrient Status Survey Report 2016 [31]. The participants were not required to provide written informed consent for the current analysis, since it was a secondary data analysis that did not involve any interaction with them.

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
