# Peer review of "The Hidden Hunger among Nepalese Non-Pregnant Women Aged 15–49 Years: The Role of Individual, Household, and Community-Level Factors"

_ijerph, 2024, doi:10.3390/ijerph21070875_

Round 1

Reviewer 1 Report

Comments and Suggestions for Authors

This paper investigates micronutrient deficiencies among non-pregnant Nepalese women aged 15-49 using data from the 2016 Nepal National Micronutrient Status Survey. It analyzes the prevalence of deficiencies in ferritin, soluble transferrin receptor, red blood cell folate, and zinc. It identifies factors such as region, body mass index, age, and household wealth that influence these deficiencies, offering insights for targeted public health interventions.

While offering valuable insights into the situation among Nepalese women, the study's findings ultimately hinder the ability to draw more concrete conclusions. To enhance clarity and maximize the utility of the provided data, please incorporate the following information:

Could you explain the methods for collecting dietary data and assessing usual food intake?

Additionally, how were the short-term benefits of supplementation considered in the analysis?

What rationale guided the decision to exclude dietary diversity information from this study?

In what manner were infection and inflammatory conditions accounted for as confounding factors?

How were the reciprocal influences between infection/inflammation and iron/zinc status addressed within the study's framework?

What additional potential confounders were considered and addressed in the study's analysis?

Regarding the study's cross-sectional design, could you elaborate on how this limitation was acknowledged and managed, particularly concerning the establishment of causation between iron, zinc, and folate deficiencies and associated factors?

Elaborate on and provide empirical support for the argument that ethnic groups are highly susceptible to deficiencies in specific micronutrients.

Support claims regarding the association between poverty and micronutrient deficiencies via the inclusion of relevant up-to-date references.

Explain the correlation observed between higher BMI and the manifestation of deficiencies in certain nutrients, offering an interpretation of the potential causes.

Could you discuss the strategies implemented to address potential recall bias stemming from the collection and analysis of self-reported data, and how this might have influenced the findings of the study?

Clearly articulate the limitations of the study and propose strategies for overcoming them.

Specify the additional data and resources required to provide the Nepalese government and other stakeholders with the necessary tools for implementing appropriate public health interventions.

Comments on the Quality of English Language

Minor editing of English language required

Author Response

Reviewer 1

Reviewer Enquiry

Researcher Responses

Could you explain the methods for collecting dietary data and assessing usual food intake?

Types of Foods Consumed by Women 15-49 years on the Preceding Day of the Survey were collected through a questionnaire.

Additionally, how were the short-term benefits of supplementation considered in the analysis?

Recent micronutrient supplementation intake (zinc, iron, folic acid, vitamin A, multiple micronutrient supplementation or powders); whether these supplementations were consumed within the last 24 hours;

Reference:

Ministry of Health Nepal, New ERA, UNICEF, EU, USAID, CDC. Nepal National Micronutrient Status Survey – 2016. Kathmandu, Nepal: Ministry of Health, Nepal, 2017

What rationale guided the decision to exclude dietary diversity information from this study?

The rationale was based on the cut-off point of  p= 0.20 is a good choice for the prognostic model and argument for the backward elimination process.

Heinze G, Dunkler D. Five myths about variable selection. Transplant International. 2017;30(1):6-10

In what manner were infection and inflammatory conditions accounted for as confounding factors?

Iron supplementation would reduce diarrhoea and justify adjusting for health status, including diarrhoea in the past 28 days and frequent diarrhea may not give your body enough time to absorb nutrients such as iron from your food.

References:

Gasche, C., Lomer, M. C. E., Cavill, I., & Weiss, G. (2004). Iron, anaemia, and inflammatory bowel diseases. Gut, 53(8), 1190-1197.

Barkhidarian B, Roldos L, Iskandar MM, Saedisomeolia A, Kubow S. Probiotic supplementation and micronutrient status in healthy subjects: A systematic review of clinical trials. Nutrients. 2021 Aug 28;13(9).

How were the reciprocal influences between infection/inflammation and iron/zinc status addressed within the study's framework?

I have added the text below to the edited manuscript.

… but this occurrence could be due to health problems, including fever, faced by rural women, and a cross-tabulation between residence and fever revealed that 9 in 10 non-pregnant women in rural areas had a fever compared to 1 in 10 in urban areas

What additional potential confounders were considered and addressed in the study's analysis?

We have addressed all potential confounders collected for women aged 15-49 years used in this dataset.

Regarding the study's cross-sectional design, could you elaborate on how this limitation was acknowledged and managed, particularly concerning the establishment of causation between iron, zinc, and folate deficiencies and associated factors? such as region, body mass index, age, and household wealth that influence these deficiencies, offering insights for targeted public health interventions.

Agreed, and we have added the text below to the manuscript.

and future research using a longitudinal design, or an interventional study is necessary to establish causality between these deficiencies (iron, zinc, and folate) and associated factors

Elaborate on and provide empirical support for the argument that ethnic groups are highly susceptible to deficiencies in specific micronutrients

A hospital-based cross-sectional study carried out in Janakpur reported that being Janajati (Indigenous Terai people), have lower education, less frequent antenatal visits, not receiving deworming medication, and having inadequate dietary diversity are found to be the significant factors (Yadav et al., 2021)

Reference:

Yadav, U. K., Ghimire, P., Amatya, A., & Lamichhane, A. (2021). Factors associated with anemia among pregnant women of underprivileged ethnic groups attending antenatal care at provincial level hospital of Province 2, Nepal. Anemia2021(1), 8847472.

Support claims regarding the association between poverty and micronutrient deficiencies via the inclusion of relevant up-to-date references.

The reference below has been added.

Bhandari, S., & Banjara, M. R. (2015). Micronutrients deficiency, a hidden hunger in Nepal: prevalence, causes, consequences, and solutions. International scholarly research notices2015(1), 276469.

Explain the correlation observed between higher BMI and the manifestation of deficiencies in certain nutrients, offering an interpretation of the potential causes.

We have added the text below:

These findings could be attributed to the fact that being overweight in non-pregnant women can lead to iron deficiency by inhibiting the absorption of dietary iron from the duodenum (Aigner et al, 2014)). This can worsen the problem of micronutrient deficiencies, especially in populations where plant-based diets are common, and the availability of dietary iron is already low (Alshwaiyat et al, 2021).

References:

Aigner, E., Feldman, A., & Datz, C. (2014). Obesity as an emerging risk factor for iron deficiency. Nutrients6(9), 3587-3600

Alshwaiyat, N. M., Ahmad, A., Wan Hassan, W. M. R., & Al‑Jamal, H. A. N. (2021). Association between obesity and iron deficiency. Experimental and Therapeutic Medicine22(5), 1-7.

Could you discuss the strategies implemented to address potential recall bias stemming from the collection and analysis of self-reported data, and how this might have influenced the findings of the study?

Agreed, and we have added the text below:

consistent measures were used for factors such as diet habits that would reduce recall bias (Rantala et al., 2022).

Reference:

Rantala, E., Balatsas-Lekkas, A., Sozer, N., & Pennanen, K. (2022). Overview of objective measurement technologies for nutrition research, food-related consumer and marketing research. Trends in Food Science & Technology125, 100-113.

Clearly articulate the limitations of the study and propose strategies for overcoming them.

Already addressed above

Specify the additional data and resources required to provide the Nepalese government and other stakeholders with the necessary tools for implementing appropriate public health interventions.

There is no additional data of this nature in Nepal, and to the best of our knowledge and due to the expensive nature of collecting and storing blood from a nationally representative survey, the first Nepal Micronutrient Status Survey (NMSS) study was conducted in 1998 and the second NMSS was conducted in 2016.

Reviewer 2 Report

Comments and Suggestions for Authors

See attached file 

Author Response

Reviewer 2

Reviewer Enquiry

Researcher Responses

I respectfully suggest the following: 1. This analysis could be enriched with some projection to more recent years

Thanks

Is this analysis currently justified based on the current situation of the study population?

Thanks

Although there is little scientific evidence with this design, how representative are these results today? This analysis corresponds to data collected since 2016.

To the best of our knowledge, the first Nepal Micronutrient Status Survey (NMSS) study was conducted in 1998, and the second NMSS was conducted in 2016. This was due to the expensive nature of collecting and storing blood from a nationally representative survey. Hence, owing to the time lag and cost of conducting another survey, this study is still very recent and would inform future public health policy.

Regarding the analysis of food consumption, Table 1 shows the percentage of the population that reported consumption; however, I suggest that if possible, and if the data are available, the analysis could be enriched by considering the values of the adequacy of consumption in the diet, or the contribution of nutritional components to illustrate the analysis.

Good point, but to our knowledge, no relevant literature supports the classification of  “adequacy of consumption in the diet” or “inadequacy of consumption in the diet”. Rather, the rationale for including food consumption variables was based on the cut-off point of p= 0.20, which is a good choice for the prognostic model and argument for the backward elimination process (Heinze &. Dunkler, 2017).

Heinze G, Dunkler D. Five myths about variable selection. Transplant International. 2017;30(1):6-10

Otherwise, the analysis of the habitual diet is poor and therefore the objective of the study may have to be reconsidered.

As indicated above, the habitual diet analysis was sound and supported by relevant literature.

The conclusions mention that certain food groups such as meat and fish were associated with the deficiencies analyzed but require further discussion due to the limited methodology used

Agreed and it was a typo error and has now been changed to “non-consumption of IFA supplements.”

Or, describe the limitations of the dietary analysis in this work

Agree, and we have added the text below to the manuscript.

consistent measures were used for factors such as diet habits that would reduce recall bias

Reference:

Rantala, E., Balatsas-Lekkas, A., Sozer, N., & Pennanen, K. (2022). Overview of objective measurement technologies for nutrition research, food-related consumer and marketing research. Trends in Food Science & Technology125, 100-113.

Reviewer 3 Report

Comments and Suggestions for Authors

Very well-written.

There are a few suggestions-

1. The abstract is structured and represents the study. However, the headings need to be included. Line 21- should be the prevalence of deficiency of or decreased levels of ferritin, … . Similarly, line 22- less prone to ferritin deficiency or Iron deficiency evaluated by ferritin levels….

2. The introduction adequately describes the existing literature and the lacunae that the research is addressing. However, a definition for the term Hidden-hunger (or explanation) needs to be included in the introduction. [i.e. Hidden hunger is the presence of multiple micronutrient....

3. Methodology is well-described and clear. The design is appropriate and provides statistical analysis in detail. However, in the outcome indicators, a reference needs to be included for each parameter and the recommended volume required. Furthermore, recent pregnancy, childbirth, or breastfeeding can be confounders of nutritional deficiencies (especially Iron). Non-pregnant women generally mean they were not pregnant at the time of study. Marriage status is considered in the study but it is not clear if the parity is taken into consideration.

4. The references are uniform, appropriate, and well-cited. However, the style may need to be modified to MDPI format.

Comments on the Quality of English Language

Formatting and language- Specific modifications required-

Pg-3 -Lines 106-109- Needs to be reformatted for clarity (An indicator of iron insufficiency when iron stores are depleted (and assuming the absence of other causes of abnormal erythropoiesis) is known as soluble transferrin receptor (sTfR), which can be elevated by thalassemia and is thought to be less influenced by inflammation and infection than ferritin). Pg 3-line 113- Iron deficiency was defined as if adjusted ferritin < 15.0 μg.

Author Response

Reviewer 3

Reviewer Enquiry

Researcher Responses

The abstract is structured and represents the study. However, the headings need to be included. Line 21- should be the prevalence of deficiency of or decreased levels of ferritin, … . Similarly, line 22- less prone to ferritin deficiency or Iron deficiency evaluated by ferritin levels….

Agreed, but we are only allowed 200 words in the abstract.

The title is appropriate.

Thanks

The introduction adequately describes the existing literature and the lacunae that the research is addressing. However, a definition for the term Hidden-hunger (or explanation) needs to be included in the introduction. [i.e. Hidden hunger is the presence of multiple micronutrient deficiencies (particularly iron, zinc, iodine, and vitamin A), which can occur without a deficit in energy intake as a result of consuming an energy-dense, but nutrient-poor diet.]

Thanks. The text below has been added to the introduction section of the manuscript.

Hidden hunger, also known as multiple micronutrient deficiencies, such as iron, zinc, iodine, and vitamin A, can occur even with an adequate energy intake due to consuming a diet that is rich in energy but lacking in nutrients (Lowe, 2021).

Lowe, N. M. (2021). The global challenge of hidden hunger: perspectives from the field. Proceedings of the Nutrition Society80(3), 283-289.

Methodology is well-described and clear. The design is appropriate and provides statistical analysis in detail. However, in the outcome indicators, a reference needs to be included for each parameter and the recommended volume required. Furthermore, recent pregnancy, childbirth, or breastfeeding can be confounders of nutritional deficiencies (especially Iron). Non-pregnant women generally mean they were not pregnant at the time of the study. Marriage status is considered in the study but it is not clear if the parity is taken into consideration.

Although not significant, we have added marital status to the manuscript.

Current pregnancy was excluded from the analysis, and as indicated in the table below, we did not include the current breastfeeding variable because about 61% were missing,

Are you currently breastfeeding your child?

n

%

Yes

595

27.8

No

234

10.9

Missing

1,314

61.3

Total

2,143

100

The results are described in detail, tables were appropriately used, are representative of the data, and clearly understandable without repetition of the data with the text

Thanks

The discussion is appropriate and explores the importance of the findings, by comparing it to the existing literature. It included the strengths and limitations. The conclusion is consistent with the results and interpretation

Thanks

Specific modifications requiredo Pg-3 -Lines 106-109- Needs to be reformatted for clarity (An indicator of iron insufficiency when iron stores are depleted (and assuming the absence of other causes of abnormal erythropoiesis) is known as soluble transferrin receptor (sTfR), which can be elevated by thalassemia and is thought to be less influenced by inflammation and infection than ferritin). Pg 3-line 113- Iron deficiency was defined as if adjusted ferritin < 15.0 μg.

Agreed and we have added the text below to the edited manuscript.

“Detailed information regarding the classification of serum zinc, sTfR, ferritin, RBC folate and Vitamin A has been described in detail elsewhere.”

Ministry of Health Nepal, New ERA, UNICEF, EU, USAID, CDC. Nepal National Micronutrient Status Survey – 2016. Kathmandu, Nepal: Ministry of Health, Nepal, 2017

Suchdev PS, Namaste S ML, Aaron GJ, Raiten DJ , Brown KH , Flores-Ayala R, on behalf of the BRINDA Working Group. 2016 Overview of the Biomarkers Reflecting Inflammation and Nutritional Determinants of Anemia (BRINDA) Project Advances in Nutrition, Volume 7, Issue 2, 1 March 2016, Pages 349–356.

The references are uniform, appropriate, and well-cited. However, the style may need to be modified to MDPI format

Thanks, and corrected.

Reviewer 4 Report

Comments and Suggestions for Authors

Dear Sir / Madam,

Please find below my comments and suggestions following the review of the study titled ‘The hidden hunger among Nepalese non-pregnant women aged 15-49 years: The role of individual, household, community-level factors’.

This study addresses an important topic concerning micronutrient deficiencies among Nepalese non-pregnant women and potential socio-ecological determinants. Nevertheless there are various aspects that could be enhanced with the presentation of the data and the facts requiring considerable revision. Suggestions to improve the manuscript have been presented below:

1.     There are numerous instances throughout the manuscript where the authors have missed out words eg: Line 206-207 “The likelihood of folate was significantly lower among non-pregnant women from the richest households compared with those from the poorest households”. The authors are requested to read through the manuscript and rectify all such deletions.

2.     Introduction:

·       I would rephrase line 57-58 as follows: Individuals with deficiencies of folate are prone to disorders such as heart disease, stroke, and depression, and women with folate deficiencies are at increased risk of giving birth to children with neural tube defects and other birth defects.

·       Line 60-61 – “causes of micronutrients [6, 20]” the authors have missed the word “deficiencies” after micronutrients.

·       The authors state their objective as examining the association of deficiencies with various factors but the ‘Introduction’ does not delineate this adequately. The authors have only described the effects of deficiencies in the ‘Introduction’ but nowhere have they discussed the association of individual or household factors as have been described in literature. The authors are requested to include in the discussion what existing literature contributes to this topic and to clearly demonstrate the need of the study with its current objectives.

3.     Materials and Methods:

·       Line 82-83: The term adolescent is used to refer to an individual up to 19 years of age – after that they are called adults. The authors are requested to rectify their classification of the sample as adolescent girls and adult non-pregnant women.

·       Line 96 – The respondents were explained not educated – Please rectify

·       The authors should provide the methods used for estimate zinc and RBT concentrations and give adequate details of the methods used including CVs of biochemical assays so that they can be replicated in future.

4.     Results:

·       In Table 1, what does the variable mean food groups mean? Also, it is indicated that the result is expressed as mean (sd) but there is no sd given.

·       Figure 1 uses the term BRINDA before the variables – The authors are requested to clarify in the methods section what this means and how they used it to adjust for the levels.  

·       Figure 1 title: The bars indicate prevalence of deficiency. The error bars indicate 95% CI.

·       Line 192 – 194: “However, non-pregnant women from the same ethnic group were significantly more likely to have iron deficiency (sTfR levels) compared with those from the Brahmi region” – what do you mean by same ethnic group?

·       In Table 3: Please give the full-forms of BIS and IFA in the legend

·       Table 3 – The authors are requested to specify what have the variables been adjusted for?

5.     Discussion:

·       The authors assessed the food group consumption. Are they able to use this data to support ethnic differences in risk for iron deficiency?

·       The authors have compared their findings with Western and High-income country populations. Weren’t studies from populations like India or Bangladesh available for comparison considering their characteristics are similar to that of the Nepalese population?

·       Findings such as ethnic differences in deficiencies, the increased susceptibility to folate deficiency among those consuming vitamin A rich foods have not been discussed in adequately – neither have the findings been interpreted nor have any speculations are made for the reason for their occurrence.

·       The finding reported in Line 291 “non-pregnant women who had a fever prior to the survey were significantly less predisposed to zinc deficiency” seems quite strange as I do not think this has been reported anywhere. I think this finding could be random or it is possible that that dehydration due to fever could be a reason for this occurrence? The authors need to discuss this finding appropriately in the light of extant literature.

Comments on the Quality of English Language

Moderate English language editing is required including rectifying all instances of deleted words.

Author Response

Reviewer 4

Reviewer Enquiry

Researcher Responses

There are numerous instances throughout the manuscript where the authors have missed out words eg: Line 206-207 “The likelihood of folate was significantly lower among non-pregnant women from the richest households compared with those from the poorest households”. The authors are requested to read through the manuscript and rectify all such deletions

Agreed, and we have replaced likelihood with odds.

I would rephrase line 57-58 as follows: Individuals with deficiencies of folate are prone to disorders such as heart disease, stroke, and depression, and women with folate deficiencies are at increased risk of giving birth to children with neural tube defects and other birth defects.

Agreed, and we have edited the text by adding the most recent reference.

Line 60-61 – “causes of micronutrients [6, 20]” the authors have missed the word “deficiencies” after micronutrients.

Thanks, and we have added the text below.

“deficiencies”

The authors state their objective as examining the association of deficiencies with various factors but the ‘Introduction’ does not delineate this adequately. The authors have only described the effects of deficiencies in the ‘Introduction’ but nowhere have they discussed the association of individual or household factors as have been described in literature. The authors are requested to include in the discussion what existing literature contributes to this topic and to clearly demonstrate the need of the study with its current objectives.

We have edited the introduction to reflect individual, household and community factors.

Line 82-83: The term adolescent is used to refer to an individual up to 19 years of age – after that they are called adults. The authors are requested to rectify their classification of the sample as adolescent girls and adult non-pregnant women.

I agree, and we have added a reference to distinguish between them.

 Line 96 – The respondents were explained not educated – Please rectify

Agreed and we have adjusted the text and now reads.

Informed consent was obtained from the respondents prior to their participation. Written informed consent was obtained from those who could who were educated and could read, while oral informed consent with the witness signatures was obtained from those who could not read, and full details regarding consent from participants were reported in detail elsewhere.

Reference:

Ministry of Health Nepal, New ERA, UNICEF, EU, USAID, CDC. Nepal National Micronutrient Status Survey – 2016. Kathmandu, Nepal: Ministry of Health, Nepal, 2017

The authors should provide the methods used for estimate zinc and RBT concentrations and give adequate details of the methods used including CVs of biochemical assays so that they can be replicated in future.

We have provided references so that the researcher can replicate the same methods.

In Table 1, what does the variable mean food groups mean? Also, it is indicated that the result is expressed as mean (sd) but there is no sd given.

Thanks for bringing this to our attention. We have added mean values, and it now reads' mean food consumed' instead of ‘food group.’

Figure 1 uses the term BRINDA before the variables – The authors are requested to clarify in the methods section what this means and how they used it to adjust for the levels.  

Thanks, and I have added the text below:

The Biomarkers Reflecting Inflammation and Nutrition Determinants of Anemia (BRINDA) working group. BRINDA working group - recommends adjusting serum zinc, sTfR and ferritin for non-pregnant women aged 15-49 years, which were used in this study

Reference:

Suchdev PS, Namaste S ML, Aaron GJ, Raiten DJ , Brown KH , Flores-Ayala R, on behalf of the BRINDA Working Group. 2016 Overview of the Biomarkers Reflecting Inflammation and Nutritional Determinants of Anemia (BRINDA) Project Advances in Nutrition, Volume 7, Issue 2, 1 March 2016, Pages 349–356.

Figure 1 title: The bars indicate prevalence of deficiency. The error bars indicate 95% CI.

It is 95% confidence intervals (CIs) and Not error bars because we did some calculation to get the CIs.

Line 192 – 194: “However, non-pregnant women from the same ethnic group were significantly more likely to have iron deficiency (sTfR levels) compared with those from the Brahmi region” – what do you mean by same ethnic group?

We put a footnote denoting other ethnic groups and indicated ‘Other Terai Caste, Newar, and Muslim’ in brackets for clarity.

In Table 3: Please give the full-forms of BIS and IFA in the legend

Thanks, and corrected

Table 3 – The authors are requested to specify what have the variables been adjusted for?

Thanks, and corrected.

The authors assessed the food group consumption. Are they able to use this data to support ethnic differences in risk for iron deficiency?

 Thanks for the suggestion and we think this is another possible research question that cannot be answered using this data.

The authors have compared their findings with populations in Western and high-income countries. Weren’t studies from populations like India or Bangladesh available for comparison considering their characteristics are similar to that of the Nepalese population?

I have cited a few more; however, due to the expensive nature, not many studies used in South Asia collect blood from households. For Example, this study was the second survey collected in Nepal since 1998.

Findings such as ethnic differences in deficiencies, the increased susceptibility to folate deficiency among those consuming vitamin A rich foods have not been discussed in adequately – neither have the findings been interpreted nor have any speculations are made for the reason for their occurrence.

Thanks, and I have added the text below to the manuscript.

Our study found a correlation between the consumption of vitamin A-rich fruits and RBC folate levels in non-pregnant women. This finding was supported by a cross-sectional study carried out in Malaysia. The study focused on the impact of 24-hour dietary recall on blood folate and showed that females reported higher folate intake than males. This finding could be attributed to their better eating habits and access to high-folate foods like fruits and vegetables.

The finding reported in Line 291 “non-pregnant women who had a fever prior to the survey were significantly less predisposed to zinc deficiency” seems quite strange as I do not think this has been reported anywhere. I think this finding could be random or it is possible that that dehydration due to fever could be a reason for this occurrence? The authors need to discuss this finding appropriately in the light of extant literature.

Agreed and we have added the text below.

… but this occurrence could be due to health problems, including fever, faced by rural women. A cross-tabulation between residence and fever revealed that 9 in 10 non-pregnant women in rural areas had a fever compared to 1 in 10 in urban areas.

Round 2

Reviewer 1 Report

Comments and Suggestions for Authors

The manuscript has been improved, but there are still some minor issues that are missing and should be addressed before accepting the paper for publication.

The methods for dietary data collection are described, but a more detailed explanation of the type of questionnaire used would be beneficial. Please specify whether 24-hour recalls or FFQs were used and describe the procedure in detail. Additionally, clarify whether only two food groups were included or if others were also considered. If only two food groups were included, please provide the rationale for this choice.

Iron supplementation would reduce diarrhea and justify adjusting for health status, including diarrhea in the past 28 days and frequent diarrhea may not give your body enough time to absorb nutrients such as iron from your food.

This is not entirely accurate. Iron supplementation can actually cause gastrointestinal side effects, including diarrhea, in some individuals. The relationship between iron supplementation and diarrhea is complex and can vary depending on the individual's health status and the form of iron supplement used. Rewrite the sentence to reflect this.

The rationale was based on the cut-off point of  p= 0.20, which is a good choice for the prognostic model and argument for the backward elimination process.

Could you please extrapolate and make sure that this information is included in the methods section.

… but this occurrence could be due to health problems, including fever, faced by rural women, and a cross-tabulation between residence and fever revealed that 9 in 10 non-pregnant women in rural areas had a fever compared to 1 in 10 in urban areas

Zinc plays a crucial role in immune function, and deficiencies can make individuals more susceptible to infections. There is no well-established or widely recognized research specifically linking having a fever before a survey with a decreased predisposition to zinc deficiency in non-pregnant women. Please either rewrite the sentence to reflect this or provide a proper reference to support your statement. In addition, it is important to emphasize that your study's findings indicate a correlation, not necessarily causation. 

Please include the references mentioned in the review report in the manuscript.

Support claims regarding the association between poverty and micronutrient deficiencies by including relevant up-to-date references.

The reference below has been added.

Bhandari, S., & Banjara, M. R. (2015). Micronutrient deficiency, a hidden hunger in Nepal: prevalence, causes, consequences, and solutions. International scholarly research notices, 2015(1), 276469.

Additional references could be added to support the statement and discussion further. Potentially consider some of the following:

Victora, C. G., Christian, P., Vidaletti, L. P., Gatica-Domínguez, G., & Menon, P. (2021). Revisiting maternal and child undernutrition in low-income and middle-income countries: variable progress towards an unfinished agenda. The Lancet, 397(10282), 1388-1399. doi: 10.1016/S0140-6736(20)32332-2

Black, R. E., Victora, C. G., Walker, S. P., Bhutta, Z. A., Christian, P., de Onis, M., ... & Uauy, R. (2013). Maternal and child undernutrition and overweight in low-income and middle-income countries. The Lancet, 382(9890), 427-451. doi: 10.1016/S0140-6736(13)60937-X

Bhutta, Z. A., Das, J. K., Rizvi, A., Gaffey, M. F., Walker, N., Horton, S., ... & Black, R. E. (2013). Evidence-based interventions for improvement of maternal and child nutrition: what can be done and at what cost? The Lancet, 382(9890), 452-477. doi: 10.1016/S0140-6736(13)60996-4

consistent measures were used for factors such as diet habits that would reduce recall bias (Rantala et al., 2022).

Please explain consistent measures in more detail, and describe them.

There is no additional data of this nature in Nepal, and to the best of our knowledge and due to the expensive nature of collecting and storing blood from a nationally representative survey, the first Nepal Micronutrient Status Survey (NMSS) study was conducted in 1998 and the second NMSS was conducted in 2016.

It's understandable, but this could serve as an opportunity to highlight the importance of further research on dietary data and the allocation of resources. This would better equip the Nepalese government and other stakeholders with the essential tools needed to implement effective public health interventions when the opportunity arises. Please include this in your paper. 

Comments on the Quality of English Language

Certain sentences should be rewritten to ensure they convey clear and valid statements, as suggested. Additional English grammar corrections could be made, and proper referencing (numbers in brackets are not always in the right positions). Please update the reference list as suggested.

Author Response

Reviewer Enquiry            

Researcher Responses

The methods for dietary data collection are described, but a more detailed explanation of the type of questionnaire used would be beneficial. Please specify whether 24-hour recalls or FFQs were used and describe the procedure in detail. Additionally, clarify whether only two food groups were included or if others were also considered. If only two food groups were included, please provide the rationale for this choice.

Response: already specific in the manuscript – see lines 104-106

Iron supplementation would reduce diarrhea and justify adjusting for health status, including diarrhea in the past 28 days and frequent diarrhea may not give your body enough time to absorb nutrients such as iron from your food.

Response: the 28 days was a typo error and has already been corrected.

This is not entirely accurate. Iron supplementation can actually cause gastrointestinal side effects, including diarrhea, in some individuals. The relationship between iron supplementation and diarrhea is complex and can vary depending on the individual's health status and the form of iron supplement used. Rewrite the sentence to reflect this.

We couldn’t locate this statement in the manuscript. Your analysis did not find a relationship between iron supplementation and diarrhoea.

Could you please extrapolate and make sure that this information is included in the methods section?

… but this occurrence could be due to health problems, including fever, faced by rural women, and a cross-tabulation between residence and fever revealed that 9 in 10 non-pregnant women in rural areas had a fever compared to 1 in 10 in urban areas

Thanks, and for clarity, we have modified the text. This additional data analysis was conducted to support our findings, which are reported in the discussion section of the manuscript. We think it should be in the discussion section because that is where we reported the findings, and it was not part of the main analysis for the study.

Support claims regarding the association between poverty and micronutrient deficiencies by including relevant up-to-date references.

Recommended References added

Please explain consistent measures in more detail, and describe them

The text has been modified and now reads:

Consistent measures such as self-reports and combinations of technologies, including various sensor- and image-based tools for detecting solid and liquid dietary consumption, could be used to reduce recall bias

There is no additional data of this nature in Nepal. To the best of our knowledge, the first Nepal Micronutrient Status Survey (NMSS) study was conducted in 1998, and the second NMSS was conducted in 2016. Due to the expensive nature of collecting and storing blood from a nationally representative survey, the first NMSS was conducted in 1998, and the second NMSS was conducted in 2016.

It's understandable, but this could serve as an opportunity to highlight the importance of further research on dietary data and the allocation of resources. This would better equip the Nepalese government and other stakeholders with the essential tools needed to implement effective public health interventions when the opportunity arises. Please include this in your paper. 

We have added the text below to the manuscript.

Furthermore, due to the high cost of collecting and storing blood for a nationally representative micronutrient survey, further research on dietary data and iron deficiencies (ferritin and sTfR concentrations) is important, and this will better prepare the Nepalese government and other stakeholders with the necessary tools to implement effective public health interventions when the opportunity arises

Certain sentences should be rewritten to ensure they convey clear and valid statements, as suggested. Additional English grammar corrections could be made, as well as proper referencing (numbers in brackets are not always in the right positions). Please update the reference list as suggested.

The English-speaking librarian has improved the English language, including grammar.

Reviewer 4 Report

Comments and Suggestions for Authors

Thank you for revising your manuscript. You have effectively addressed the reviewers' comments, significantly improving the clarity and quality of the work.

Comments on the Quality of English Language

Please recheck the manuscript for grammatical and typographical errors.

Author Response

The English-speaking librarian has improved the English language, including grammar.